# Neurovascular anatomy of dwarf dinosaur implies precociality in sauropods

**Marco Schade[1,2]\*, Nils Knötschke[3], Marie K Hörnig[2], Carina Paetzel[2], Sebastian Stumpf[4]**

[1]University of Greifswald, Institute of Geography and Geology, Palaeontology and Historical, Greifswald, Germany; [2]University of Greifswald, Zoological Institute and Museum, Cytology and Evolutionary Biology, Greifswald, Germany; [3]Mineralientage, Oberhaching, Germany; [4]University of Vienna, Department of Palaeontology, Vienna, Austria

**Abstract** Macronaria, a group of mostly colossal sauropod dinosaurs, comprised the largest terrestrial vertebrates of Earth's history. However, some of the smallest sauropods belong to this group as well. The Late Jurassic macronarian island dwarf *Europasaurus holgeri* is one of the most peculiar and best-studied sauropods worldwide. So far, the braincase material of this taxon from Germany pended greater attention. With the aid of micro-computed tomography (microCT), we report on the neuroanatomy of the nearly complete braincase of an adult individual, as well as the inner ears (endosseous labyrinths) of one other adult and several juveniles (the latter also containing novel vascular cavities). The presence of large and morphologically adult inner ears in juvenile material suggests precociality. Our findings add to the diversity of neurovascular anatomy in sauropod braincases and buttress the perception of sauropods as fast-growing and autonomous giants with manifold facets of reproductive and social behaviour. This suggests that – apart from sheer size – little separated *Europasaurus* from its large-bodied relatives.

**\*For correspondence:** marco.schade@stud.uni-greifswald.de

## Editor's evaluation

The authors provide the first detailed description of the neuroanatomy of the remarkable dwarf sauropod *Europasaurus* from the Jurassic of Germany, which, at least in this regard, was not very different from some of its much larger relatives. The available evidence is compelling and convincing. The comparative sections of the manuscript are solid and provide a relatively broad overview. Based on remains of different individuals and growth stages, the authors suggest that *Europasaurus* was likely precocial. The authors also assess the likely auditory capabilities and their relevance to the reproductive and social behaviour of this island-dwelling dinosaur.

## Introduction

Sauropoda is a taxon of saurischian dinosaurs and comprise popular taxa like *Diplodocus*, *Giraffatitan*, and *Argentinosaurus* (e.g., **Bates et al., 2016**). Sauropods were taxonomically diverse and had a worldwide distribution (e.g., **Bates et al., 2016**; **Pol et al., 2021b**). Sauropods likely originated in the Late Triassic (e.g., **Rauhut et al., 2020**; **Pol et al., 2021b**) and their geologically youngest representatives vanished during the end-Cretaceous mass extinction event (e.g., **Curry Rogers and Forster, 2001**; **Bates et al., 2016**). Whereas bipedal early sauropodomorphs were probably capable of swiftly tracking down prey (**Müller et al., 2021**), the later evolutionary history of the group is characterized by an unrivaled increase in body size (among land-dwelling vertebrates), accompanied with herbivory,

**eLife digest** Dinosaurs, like all animals with spines, had their main sensory organs – the organs that allowed them to listen, taste, see, smell, think and even keep their balance – on their heads. This means that studying their fossilized skulls can provide a wealth of information about how these animals perceived their environment through so-called 'endocasts' (digital models of the cavities within the skull).

Endocasts of the skulls of many different dinosaur species already exist, but a small species called *Europasaurus holgeri* had so far not received this treatment. This sauropod lived in what is now northern Germany during the Late Jurassic period (154 million years ago), and it owed its reduced size to having become isolated on an island, where it became smaller after many generations. Schade et al. wanted to gain a better understanding of certain lifestyle aspects of the biology of *E. holgeri*, and to be able to compare the endocast anatomy of this species to other dinosaurs. To do this, the team studied the braincases of both very young and mature *E. holgeri* individuals using a technique called computer tomography.

The approach taken by Schade et al. allowed them to examine and describe in detail the inner cavities that once contained the brain, inner ears, nerves and blood supply of eight different *E. holgeri* individuals. They found that the inner ears of small and young *E. holgeri* individuals were almost as large as those of their adult counterparts, and very similar in shape. Given that inner ears have roles in both audition and the sense of equilibrium, this suggests that *E. holgeri* babies were able to leave their nest very soon after hatching. This makes it likely that the babies of the species were highly developed when they hatched, and could probably feed themselves almost immediately, possibly similar to chickens. Furthermore, the relatively large size of the part of the inner ear responsible for hearing hints at *E. holgeri* being well able to communicate with other members of the species using sound.

The findings of Schade et al. add to the diversity of the record on the anatomy of the braincases of dinosaurs. Additionally, the results support the idea that sauropods may have been herd-living animals with social interactions that grew very fast and had to be light on their feet very early in life. Finally, comparing the endocasts of *E. holgeri* to those of other dinosaurs suggests that, beyond a discrepancy in body size, this species was very similar to its larger relatives on the Jurassic mainland.

an extreme elongation in neck length and graviportal quadrupedality (e.g., *Sander et al., 2011*; *Bates et al., 2016*; *Bronzati et al., 2018*).

While fossil braincases are generally rare, studies of sauropod endocrania are nevertheless numerous (e.g., *Janensch, 1935*; *Paulina-Carabajal, 2012*; *Knoll et al., 2015*), serving as a good base for comparisons. Potentially, aspects of lifestyle can be inferred from morphological details of cavities that once contained the brain, inner ear, and other associated neurovascular structures within the bony braincase of fossil vertebrates (e.g., *Neenan et al., 2017*; *Schwab et al., 2020*; *Schwab et al., 2021*; *Ezcurra et al., 2020*; *Hanson et al., 2021*; *Choiniere et al., 2021*; however, see also *Benson et al., 2017*; *Evers et al., 2019*; *Bronzati et al., 2021*; *David et al., 2022*). Furthermore, ontogenetically induced morphological shifts of neuroanatomy can hint towards different ecological tendencies within a species, for example, in respect to bipedal or quadrupedal locomotion (*Bullar et al., 2019*).

The middle Kimmeridgian (Late Jurassic) sauropod *Europasaurus* (represented by a single species, *E. holgeri*) is regarded as an unequivocal example of insular dwarfism (although, see *Lokatis and Jeschke, 2018* for a critical view on the concept of the island rule) with paedomorphic features, having reached adult body lengths of nearly 6 m and weighing about 800 kg (*Sander et al., 2006*; *Stein et al., 2010*; *Carballido and Sander, 2013*; *Marpmann et al., 2014*). From this taxon, a great number of cranial and postcranial fossil bones are known (housed in the Dinosaurier-Freilichtmuseum Münchehagen/Verein zur Förderung der Niedersächsischen Paläontologie e.V., Rehburg-Loccum, Münchehagen, Germany; DFMMh/FV), of which the latter hint to at least 21 individuals of different ontogenetic stages (*Scheil et al., 2018*). The fossils come from shallow-marine carbonate rocks of the Langenberg quarry, assigned to the Süntel Formation, having formed in the Lower Saxony basin (see *Zuo et al., 2018*).

The paratype specimen of *Europasaurus*, DFMMh/FV 581.1, comprises a largely complete, articulated and probably mature braincase, with DFMMh/FV 581.2 and 3 representing the respective detached parietals (*Figures 1–3*; *Figure 1—figure supplements 1–4*). The outer morphology of this material has previously been described (*Marpmann et al., 2014*). For this study, the parietals were rearticulated with the preserved neurocranium and subsequently documented with micro-computed tomography (microCT). The endocranial cavities which once housed the brain, inner ears, and other soft neuroanatomical structures, such as nerves and blood supply, were then manually segmented. The articulated specimens DFMMh/FV 581.1, 2, and 3 measure about 120 mm mediolaterally, 80 mm anteroposteriorly, and 100 mm dorsoventrally.

Additionally, the specimens DFMMh/FV 1077 (*Figure 4*; *Figure 4—figure supplements 1 and 2*; adult fragmentary braincase, complete endosseous labyrinth), DFMMh/FV 466+205 (*Figures 5 and 6*; *Figure 5—figure supplement 1*; *Figure 7—figure supplements 1 and 2*; *Figure 8—figure supplements 1 and 2*; juvenile prootic and otoccipital, nearly complete endosseous labyrinth; the common bond of these two specimens has not been recognized in former studies; *Marpmann et al., 2014*), DFMMh/FV 964 and DFMMh/FV 561 (*Figure 7*; *Figure 7—figure supplements 1 and 2*; prootics of uncertain maturity, anterior labyrinth), DFMMh/FV 981.2, DFMMh/FV 898, and DFMMh/FV 249 (*Figure 8*; *Figure 8—figure supplements 1 and 2*; juvenile otoccipitals, posterior labyrinth) were documented with microCT. Since the isolated specimens contain different parts of the endosseous labyrinths, cranial nerves and vascular cavities, the respective digital models were reconstructed in order to describe, compare, and contextualize their characteristics. Whereas the smallest of these specimens (DFMMh/FV 898) hints to an approximate posterior skull width of under 50 mm, the largest specimens DFMMh/FV 581.1 and DFMMh/FV 1077 suggest a mediolateral width of about 140 mm.

The microCT data and our digital reconstructions (Europasaurus holgeri - neuroanatomy - DFMMh/FV - Schade et al. 2023 // MorphoSource) of different *Europasaurus* individuals add to the knowledge of diversity of dinosaur neuroanatomy and allow a better understanding of ontogenetic development. We discuss our findings in context of insights into the lifestyle of this long-necked insular dwarf from the Late Jurassic of Germany.

## Results

### Cranial endocast, innervation, and blood supply

As is generally the case in non-maniraptoriform dinosaurs (e.g., *Witmer and Ridgely, 2008a*; *Witmer and Ridgely, 2009*; *Knoll et al., 2015*; *Knoll et al., 2021*), many characteristics of the mid- and hindbrain are not perceivable with certainty (however, see *Evans, 2005*; *Morhardt, 2016*; *Fabbri et al., 2017*) on the braincase endocast of DFMMh/FV 581.1 (*Figure 1A*), which implies scarce correlation of the actual brain and the inner surface of the endocranial cavity (see *Watanabe et al., 2019*, for ontogenetic variations in recent archosaurs).

This endocast suggests low angles in the cerebral and pontine flexures. There is a prominent dorsal expansion, spanning from around the posterodorsal skull roof to approximately the anteroposterior mid-length of the endocast (*Figures 1 and 2*). In posterior view, the dorsal expansion is T-shaped with a more or less straight top and dorsolateral beams that become dorsoventrally higher and gradually lead over anteriorly to the area where the posterior part of the cerebral hemispheres are expected. In lateral view, the posterior-most extent of the dorsal expansion is separated from the dorsal margin of the medulla oblongata by a concavity. Anterolaterally to this concavity, the eminence for the vena capitis media is present. Although the respective openings are identifiable on DFMMh/FV 581.1 (close to a kink on the posterodorsal contact between the parietals and the supraoccipital, called 'external occipital fenestra for the caudal middle cerebral vein' in *Marpmann et al., 2014*), only an approximate reconstruction of the course of the veins was possible (due to low contrast in the microCT data; *Figures 1 and 3*). There is a large semicircular depression on the posterodorsolateral aspect of the endocast, being anterodorsally bordered by the dorsal expansion and anteroventrally by the eminence of the vena capitis media. On the anterodorsal skull roof, a mediolateral expansion of the endocast possibly marks the position of the cerebral hemisphere (*Figures 2 and 3*). In lateral view, there is a distinct ventral step on top of the endocast (also present in many other sauropod taxa; see, e.g., *Knoll and Schwarz-Wings, 2009*; *Paulina-Carabajal, 2012*; *Knoll et al., 2013*), between the anterior-most part of the dorsal expansion and the posterior part of the cerebral hemispheres,

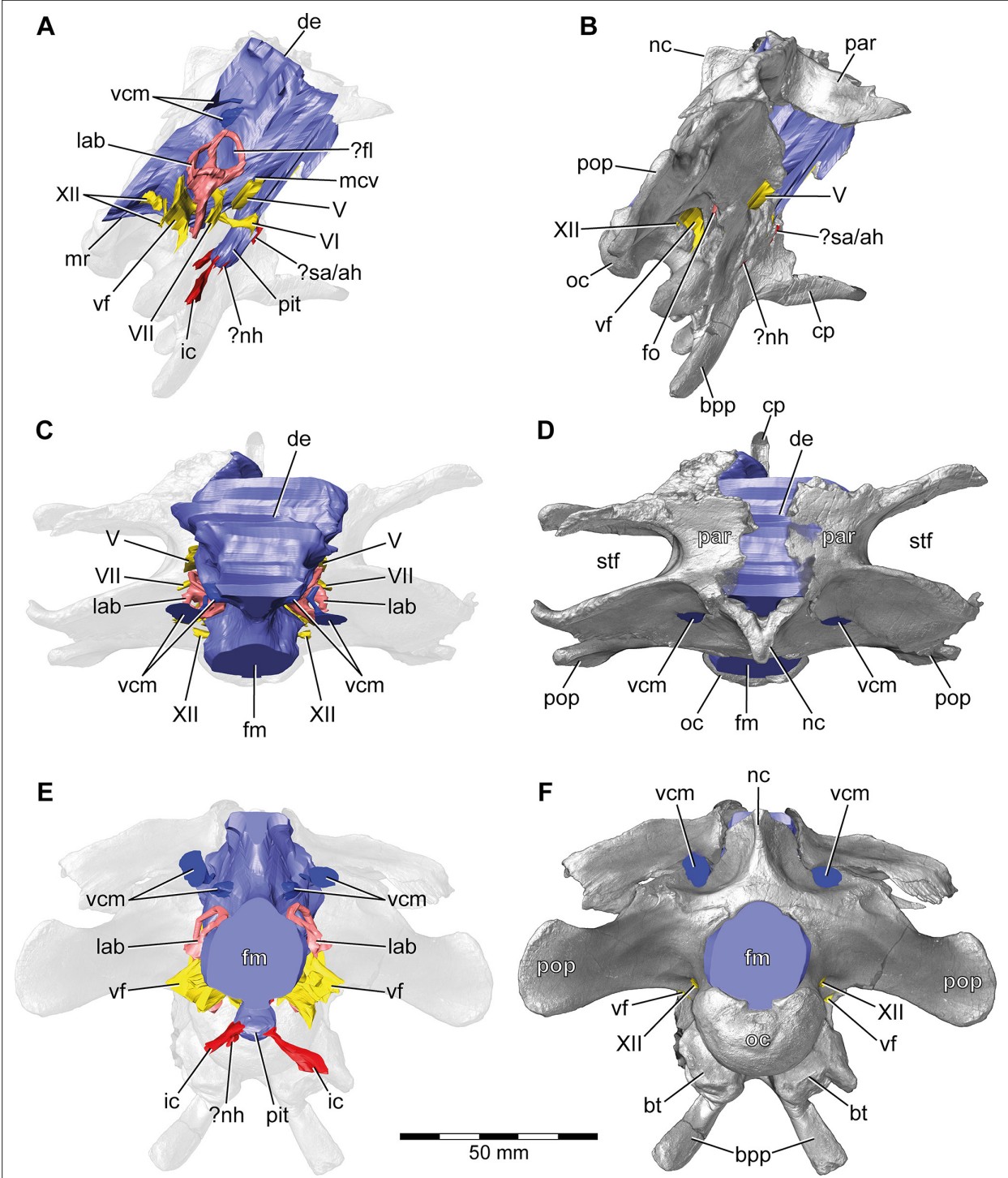

**Figure 1.** *Europasaurus holgeri*, 3D model of the braincase endocast with endosseous labyrinths and neurovascular canals of DFMMh/FV 581.1, 2, and 3 with transparent (**A,C,E**) and covering (**B,D,F**) volume rendering of the bony braincase in (**A,B**) right lateral, (**C,D**) dorsal, and (**E,F**) posterior view. Note that scale mainly applies to posterior perspective (**E,F**).?fl, potential floccular recess; ?nh, potential canal for the neurohypophysis; ?sa/ah, potential sphenoidal artery/canal for the adenohypophysis; bpp, basipterygoid process; bt, basal tuber; cp, cultriform process; de, dorsal expansion; ic, internal carotid; fm, foramen magnum; fo, fenestra ovalis; lab, endosseous labyrinth; mcv, mid cerebral vein; mr, median ridge; nc, sagittal nuchal crest; oc, occipital condyle; par, parietal; pit, pituitary; pop, paroccipital process; stf, supratemporal fenestra; vcm, vena capitis media; vf, vagal foramen; V, trigeminal nerve; VI, abducens nerve; VII, facial nerve; XII, hypoglossal nerve.

The online version of this article includes the following figure supplement(s) for figure 1:

*Figure 1 continued on next page*

*Figure 1 continued*

**Figure supplement 1.** *Europasaurus holgeri*, close-up of left lateral aspect of DFMMh/FV 581.1.

**Figure supplement 2.** *Europasaurus holgeri*, close-up of anterior endocranial floor of DFMMh/FV 581.1 in dorsal view.

**Figure supplement 3.** *Europasaurus holgeri*, close-up of 3D model of anterior endocranial floor of DFMMh/FV 581.1 in dorsal view.

**Figure supplement 4.** *Europasaurus holgeri*, close-up of right posterior endocranial wall of DFMMh/FV 581.1, viewed through the foramen magnum.

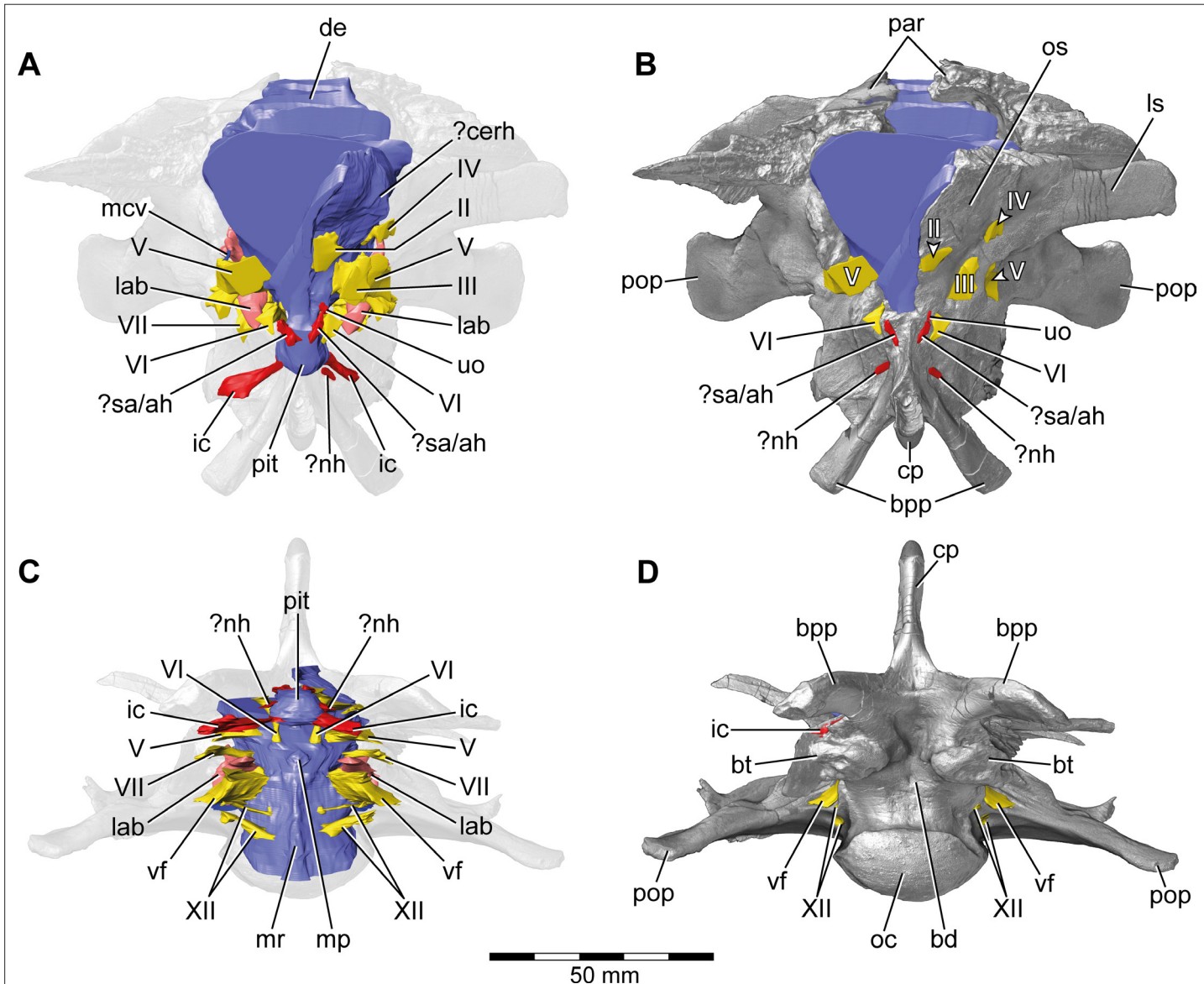

**Figure 2.** *Europasaurus holgeri*, 3D model of the braincase endocast with endosseous labyrinths and neurovascular canals of DFMMh/FV 581.1, 2, and 3 with transparent (**A,C**) and covering (**B,D**) volume rendering of the bony braincase in (**A,B**) anterior (**C,D**) and ventral view. Note that scale mainly applies to ventral perspective (**C,D**). ?cerh, potential cerebral hemisphere; ?nh, potential canal for the neurohypophysis; ?sa/ah, potential sphenoidal artery/canal for the adenohypophysis; bd, blind depression; bpp, basipterygoid process; bt, basal tuber; cp, cultriform process; de, dorsal expansion; ic, internal carotid; fm, foramen magnum; lab, endosseous labyrinth; ls; laterosphenoid; mcv, mid cerebral vein; mp, median protuberance; mr, median ridge; oc, occipital condyle; os, orbitosphenoid; par, parietal; pit, pituitary; pop, paroccipital process; uo, unclear opening; vf, vagal foramen; II, optic nerve; III, oculomotor nerve; IV, trochlear nerve; V, trigeminal nerve; VI, abducens nerve; VII, facial nerve; XII, hypoglossal nerve.

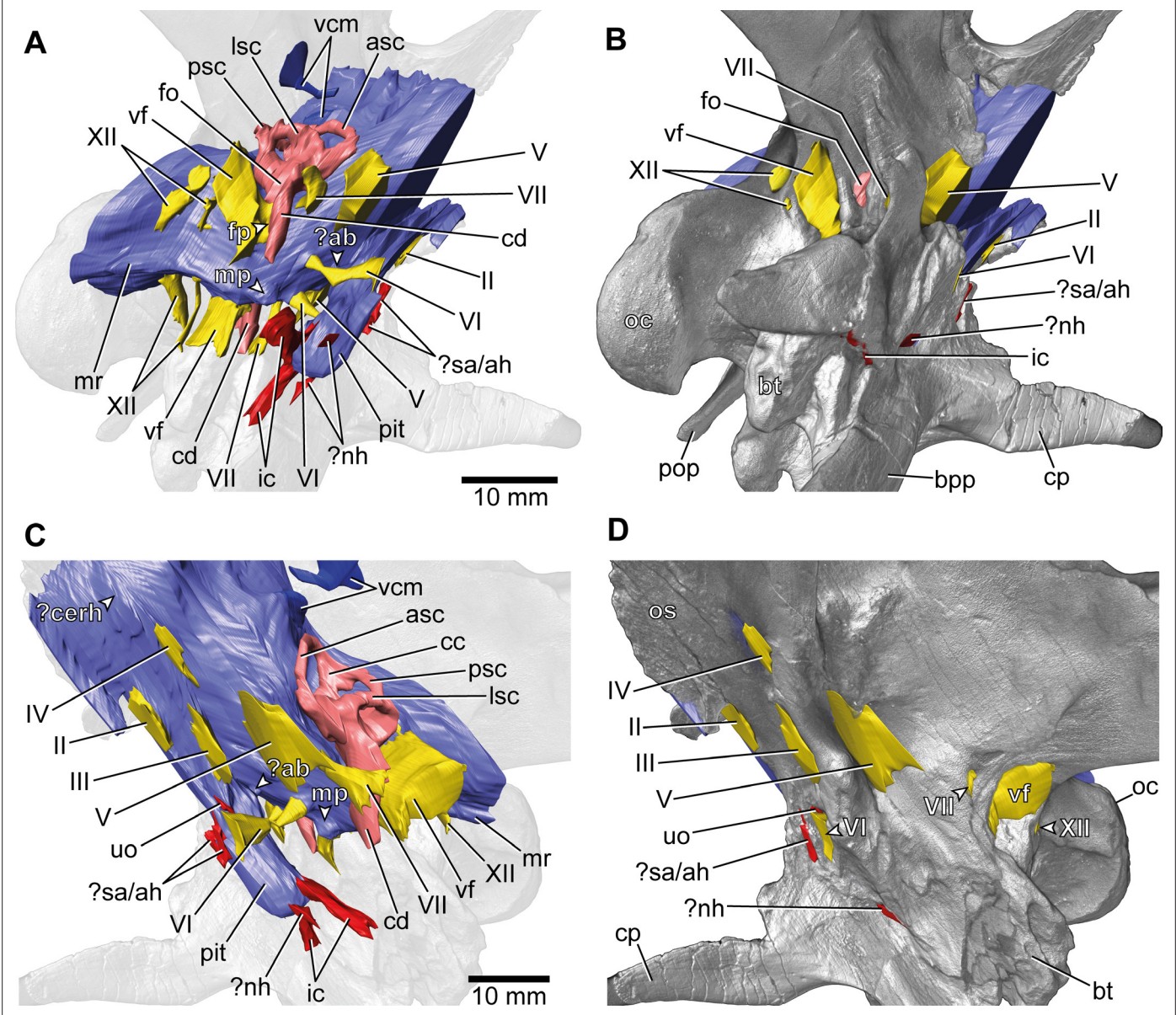

**Figure 3.** *Europasaurus holgeri*, 3D model of the braincase endocast with endosseous labyrinths and neurovascular canals of DFMMh/FV 581.1, 2, and 3 with transparent (**A,C**) and covering (**B,D**) volume rendering of the bony braincase in (**A,B**) right ventrolateral and (**C,D**) left lateral view. ?ab, potential basilar artery; ?cerh, potential cerebral hemisphere; ?nh, potential canal for the neurohypophysis; ?sa/ah, potential sphenoidal artery/canal for the adenohypophysis; bpp, basipterygoid process; bt, basal tuber; cp, cultriform process; ic, internal carotid; fo, fenestra ovalis; fp, fenestra pseudorotunda; mp, median protuberance; mr, median ridge; oc, occipital condyle; os, orbitosphenoid; pit, pituitary; pop, paroccipital process; uo, unclear opening; vcm, vena capitis media; vf, vagal foramen; II, optic nerve; III, oculomotor nerve; IV, trochlear nerve; V, trigeminal nerve; VI, abducens nerve; VII, facial nerve; XII, hypoglossal nerve.

followed by a slight ascent in anterior direction. The left side of the endocast suggests that the cerebral hemisphere impressions are delimited approximately by the contact between the orbitosphenoid and the laterosphenoid anteriorly, and by the trochlear nerve (CN IV) ventrally. Anteriorly, the orbitosphenoid bears a prominent medial incision for the optic nerve (CN II). Posteroventrally to the optic nerve canal and anteroventrally to the trochlear nerve canal, the canal of the oculomotor nerve (CN III) is situated. On the anteroventral aspect of the endocast, the pituitary reaches slightly-more ventrally than the ventral-most margin of the medulla oblongata, producing an angle of about 50° to the lateral semicircular canal (LSC) of the endosseous labyrinth (see *Paulina-Carabajal et al., 2020*). On the anterodorsal aspect of the pituitary, two small and dorsolaterally diverging canals of

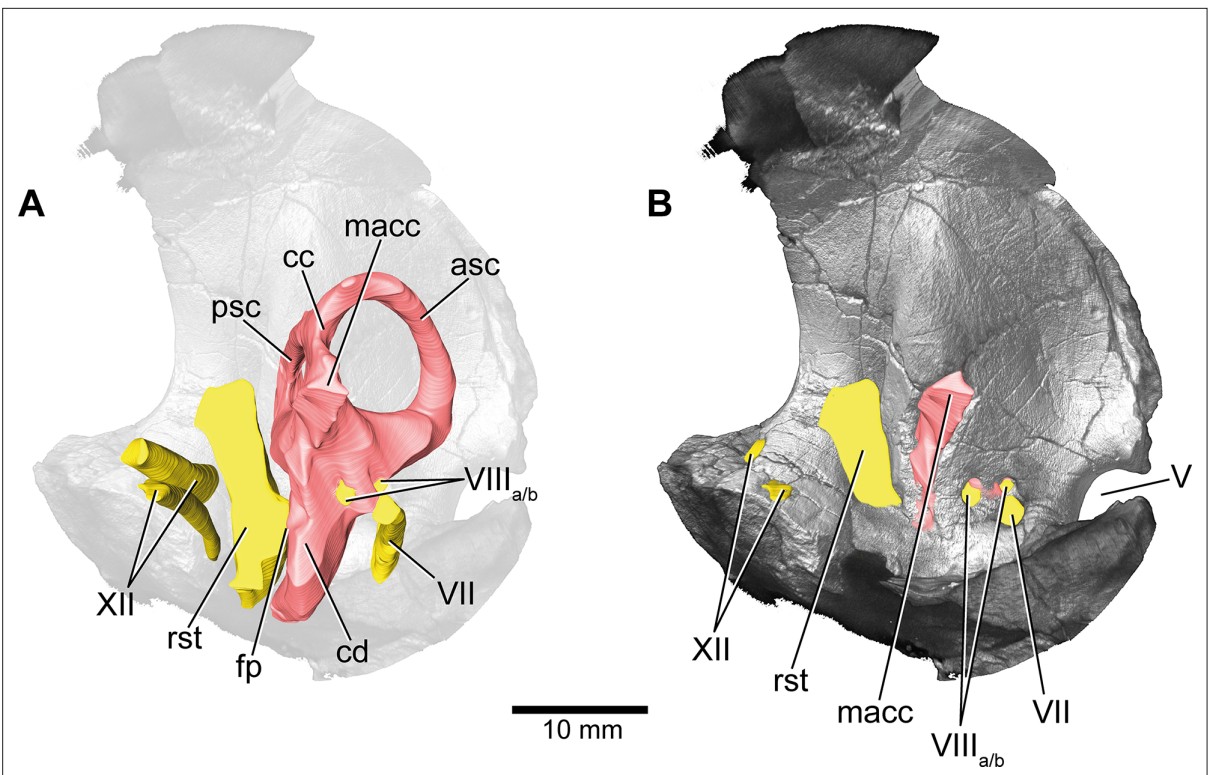

**Figure 4.** *Europasaurus holgeri*, 3D model of the left endosseous labyrinth region in DFMMh/FV 1077 with transparent (**A**) and covering (**B**) volume rendering of the bony braincase in medial view. asc, anterior semicircular canal; cc, common crus; cd, cochlear duct; fp, fenestra pseudorotunda; lsc, lateral semicircular canal; macc, medial aspect of common crus; psc, posterior semicircular canal; rst, recessus scalae tympani; V, trigeminal nerve opening; VII, facial nerve; VIIIa/b, both branches of the vestibulocochlear nerve; XII, hypoglossal nerve.

The online version of this article includes the following figure supplement(s) for figure 4:

**Figure supplement 1.** *Europasaurus holgeri*, fragmentary braincase DFMMh/FV 1077 in (**A**) ventral and (**B**) posterior view.

**Figure supplement 2.** *Europasaurus holgeri*, close-up of medial aspect of the fragmentary braincase DFMMh/FV 1077.

uncertain identity branch off (*Figures 1–3*; *Marpmann et al., 2014* labelled the openings as carotid artery: Figure 13D). In the titanosaur specimen CCMGE 628/12457 and *Sarmientosaurus*, structures of a similar position were identified as sphenoidal arteries (*Sues et al., 2015*; *Martínez et al., 2016*). However, in *Bonatitan* and the titanosaur braincase MPCA-PV-80, anterolateral openings on the pituitary, close to the abducens nerve (CN VI) canal, have been assigned to canals leading to the adenohypophysis (*Paulina-Carabajal, 2012*). Posterolaterally to these canals, the abducens nerve (CN VI) canals trend in an anteroposterior direction (*Figures 1–3*; *Figure 1—figure supplements 1–3*). The specimen DFMMh/FV 581.1 suggests a natural connection between the pituitary fossa and the left CN VI canal, close to its anterior opening. However, this condition may be due to breakage, since the microCT data suggests a continuous wall on the right side. In ventrolateral view, the left side of DFMMh/FV 581.1 shows an additional small medial opening dorsally within the depression for CN VI (*Figures 2A, B and 3C, D*; *Figure 1—figure supplements 1–3*). Because of its smooth curvature, this opening seems natural, but for preservational reasons this is not visible on the right side of the specimen. On the ventrolateral part of the pituitary, two short canals of uncertain identity are branching off ventrolaterally (*Figures 1–3*; *Figure 1—figure supplement 1*; in *Bonatitan*, anterolateral canals on the ventral portion of the pituitary have been identified as leading to the neurohypophysis; *Paulina-Carabajal, 2012*). Directly behind, the pituitary bears the long internal carotid canals, branching off ventrolaterally as well.

The endosseous labyrinth is situated within an anteroventrally inclined lateral depression of the endocast, directly ventral to the vena capitis media eminence. Here, an opening is present, leading to the medial aspect of the common crus in DFMMh/FV 581.1 and 1077 (the opening is considerably larger in the latter specimen; *Figures 4 and 6*; *Figure 1—figure supplement 4*; *Figure 4—figure*

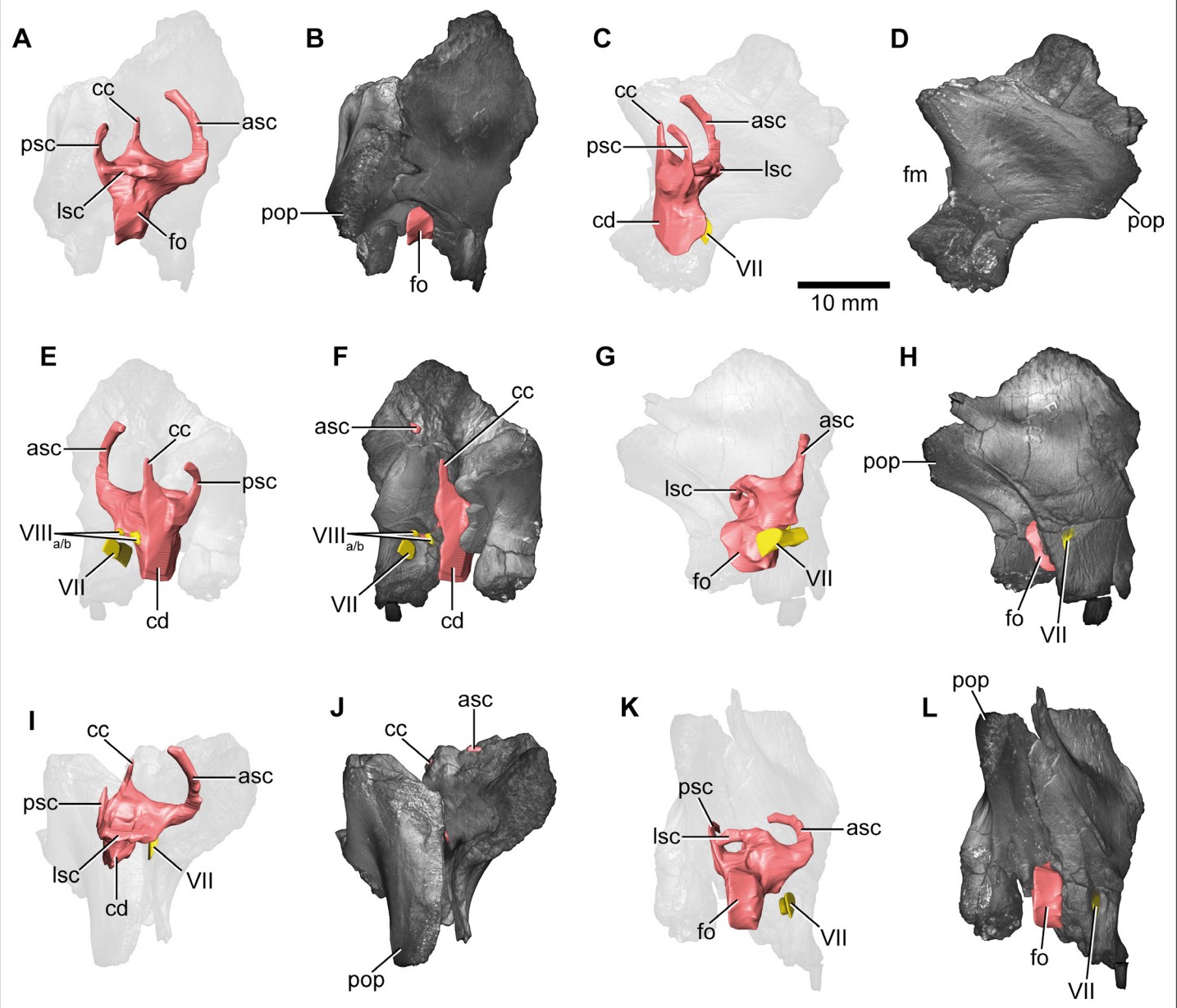

**Figure 5.** *Europasaurus holgeri*, 3D model of the right endosseous labyrinth in DFMMh/FV 466+205 with transparent (**A,C,E,G,I,K**) and covering (**B,D,F,H,J,L**) volume rendering of the bony braincase remains in (**A,B**) lateral, (**C,D**) posterior, (**E,F**) medial, (**G,H**) anterolateroventral, (**I,J**) dorsolateral, and (**K,L**) lateroventral view; in respect to the endosseous labyrinth. Note that scale mainly applies to posterior perspective (**C,D**), and that VII and VIIIa/b are not shown in (**A**) and (**B**). asc, anterior semicircular canal; cc, common crus; cd, cochlear duct; fm, foramen magnum; fo, fenestra ovalis; lsc, lateral semicircular canal; pop, paroccipital process; psc, posterior semicircular canal; VII, facial nerve; VIIIa/b, both branches of the vestibulocochlear nerve.

The online version of this article includes the following figure supplement(s) for figure 5:

**Figure supplement 1.** *Europasaurus holgeri*, isolated otoccipital (DFMMh/FV 205; **A,B**) and prootic (DFMMh/FV 466; **C,D**) in (**A**) posterior, (**B**) anterior, (**C**) lateral, and (**D**) medial view; prootic and otoccipital conjoined in (**E**) posterolateral, (**F**) lateral, (**G**) medial, (**H**) dorsal, and (**I**) ventral view.

*supplement 2*). Whereas the trigeminal (CN V), facial (CN VII), and vestibulocochlear (CN VIII; two openings) nerve canals are mainly anterior to the endosseous labyrinth, the vagal foramen (=jugular foramen for CN IX-XI and jugular vein) and two canals for the hypoglossal nerves are situated posterior to the cochlear duct (*Figures 1–3*). Within the depression for CN V, dorsally, a very small opening for the mid-cerebral vein is situated on both sides of DFMMh/FV 581.1. However, only the right canal could approximately be reconstructed (*Figures 1A and 2A*). Dorsal to the right slit-like opening for CN VII, a small depression is present in DFMMh/FV 581.1. The microCT data do not suggest penetration.

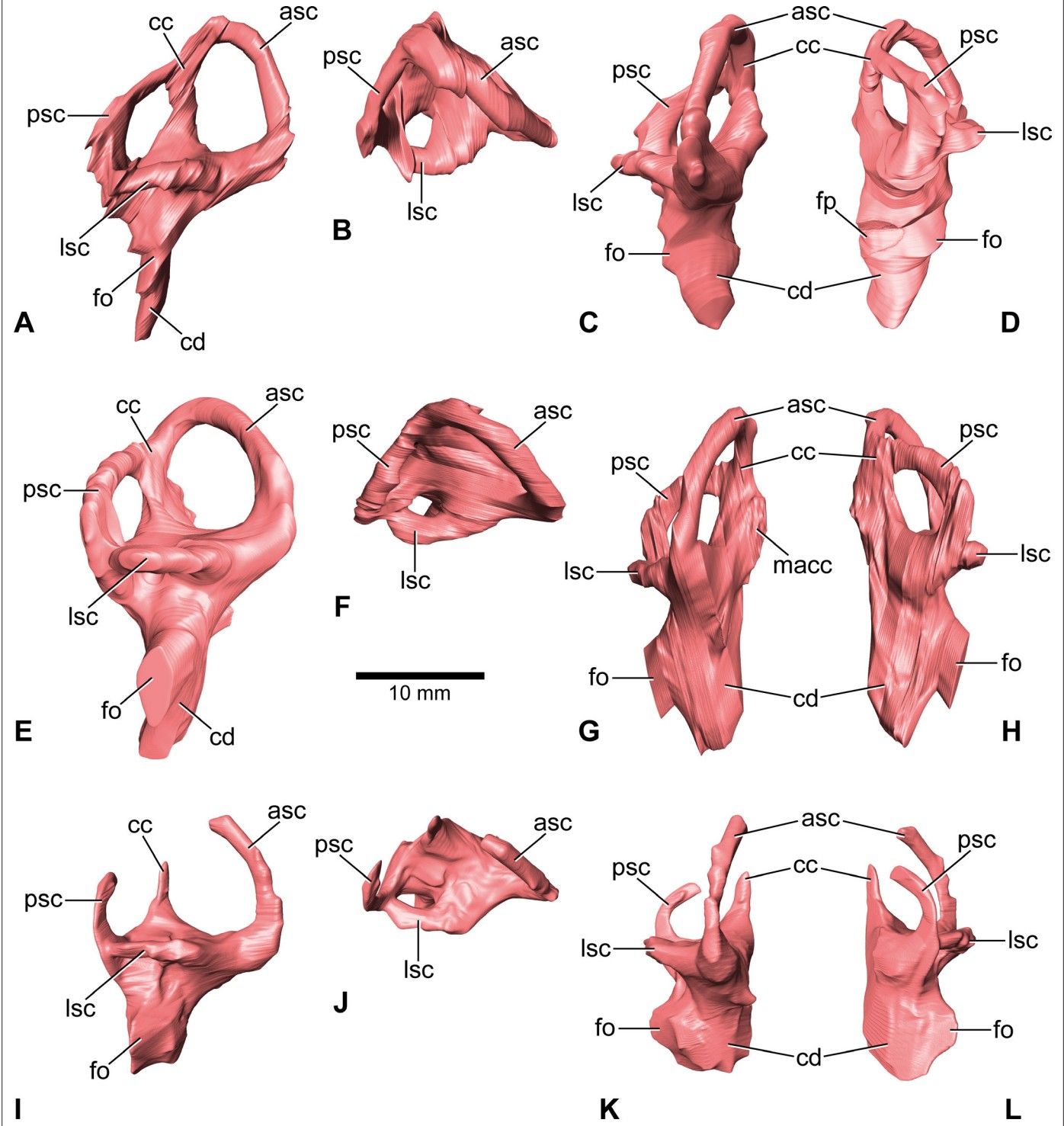

**Figure 6.** *Europasaurus holgeri*, 3D models of the endosseous labyrinth of DFMMh/FV 581.1 (**A–D**), DFMMh/FV 1077 (**E–H**; note that this model is mirrored) and DFMMh/FV 466+205 (**I–L**) in (**A,E,I**) lateral, (**B,F,J**) dorsal, (**C,G,K**), anterior and (**D,H,L**) posterior view. Note that scale mainly applies to dorsal perspective (**B,F,J**). asc, anterior semicircular canal; cc, common crus; cd, cochlear duct; fo, fenestra ovalis; fp, fenestra pseudorotunda; lsc, lateral semicircular canal; macc, medial aspect of common crus; psc, posterior semicircular canal.

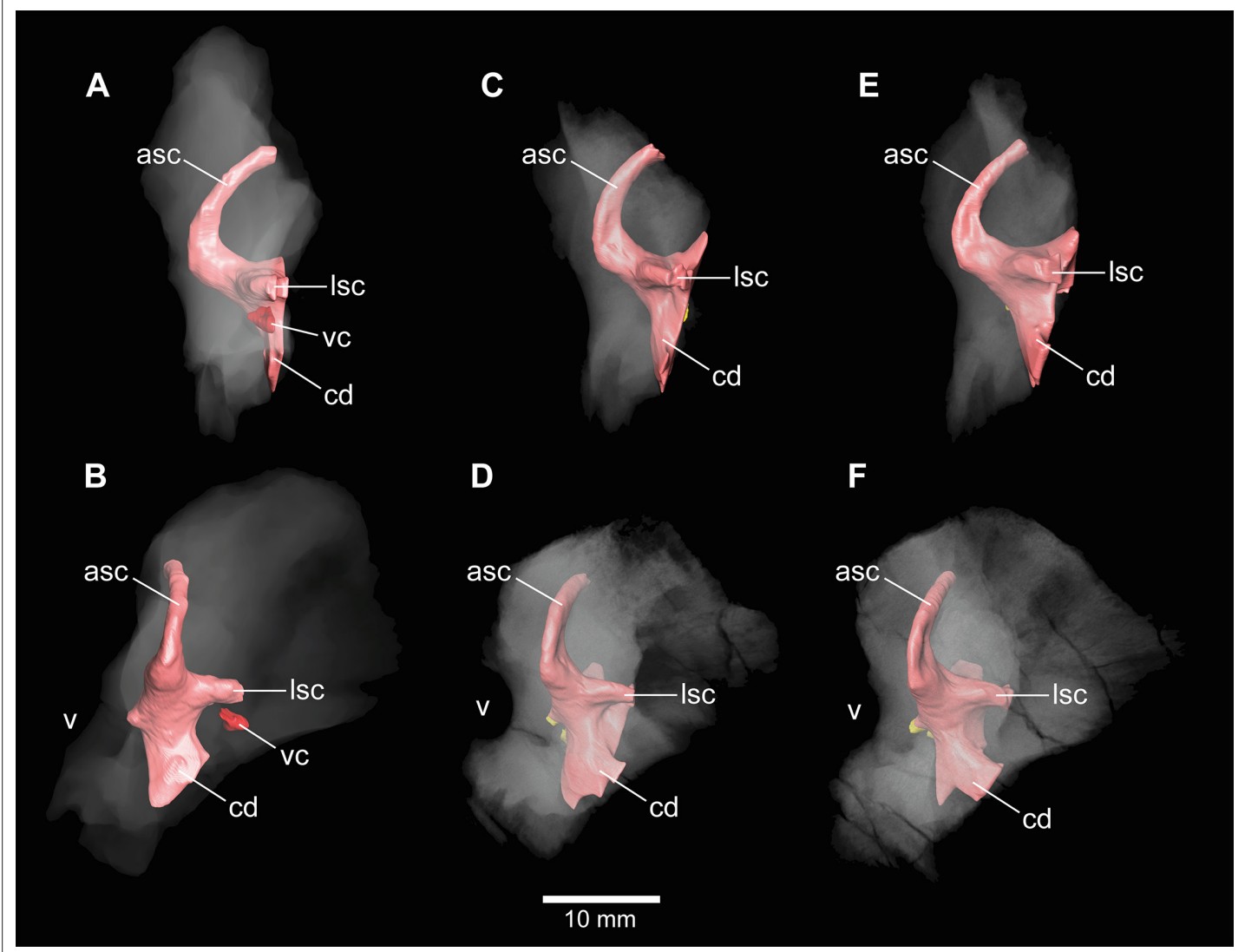

**Figure 7.** *Europasaurus holgeri*, 3D models of the anterior portions of the endosseous labyrinth in (**A**,**B**; note that this model is mirrored) DFMMh/FV 466, (**C**,**D**) DFMMh/FV 561 and (**E**,**F**) DFMMh/FV 964 in (**A**,**C**,**E**) lateral and (**B**,**D**,**F**) anterolateral view; in respect to the endosseous labyrinth. Note that scale mainly applies to anterolateral perspective (**B**,**D**,**F**). asc, anterior semicircular canal; cd, cochlear duct; lsc, lateral semicircular canal; vc, vascular cavity; V, trigeminal nerve opening.

The online version of this article includes the following figure supplement(s) for figure 7:

**Figure supplement 1.** *Europasaurus holgeri*, isolated prootics (DFMMh/FV 466, **A**,**B**; DFMMh/FV 964, **C**,**D**; DFMMh/FV 561, **E**,**F**) in (**A**,**C**,**E**) lateral and (**B**,**D**,**F**) medial view.

**Figure supplement 2.** *Europasaurus holgeri*, 3D models of isolated prootics and inner features (DFMMh/FV 466, **A**; DFMMh/FV 561, **B**; DFMMh/FV 964, **C**) in (**A**–**C**) medial view.

Whereas the posterior canals for the hypoglossal nerve (CN XII) are clearly discernable in the microCT data, the anterior ones are not as obvious to detect. However, because of the expression of their respective openings on the actual fossil, their course could be established. *Marpmann et al., 2014*, only identified one hypoglossal canal (CN XII). However, the specimens considered herein support the presence of two openings on each side. Furthermore, anterior to the proximal openings of the anterior CN XII canals, one depression each is visible in DFMMh/FV 581.1, however, the microCT data do not suggest a penetration. Anterodorsally to the endosseous labyrinth, the cerebellum appears as a mediolaterally expanded part of the endocast, almost reaching the trigeminal nerve (CN V) anteriorly and being delimited by the eminence of the vena capitis media posterodorsally (*Figure 1A*). Furthermore, a small floccular recess is present close to the mid-length of the anterior semicircular canal (ASC)

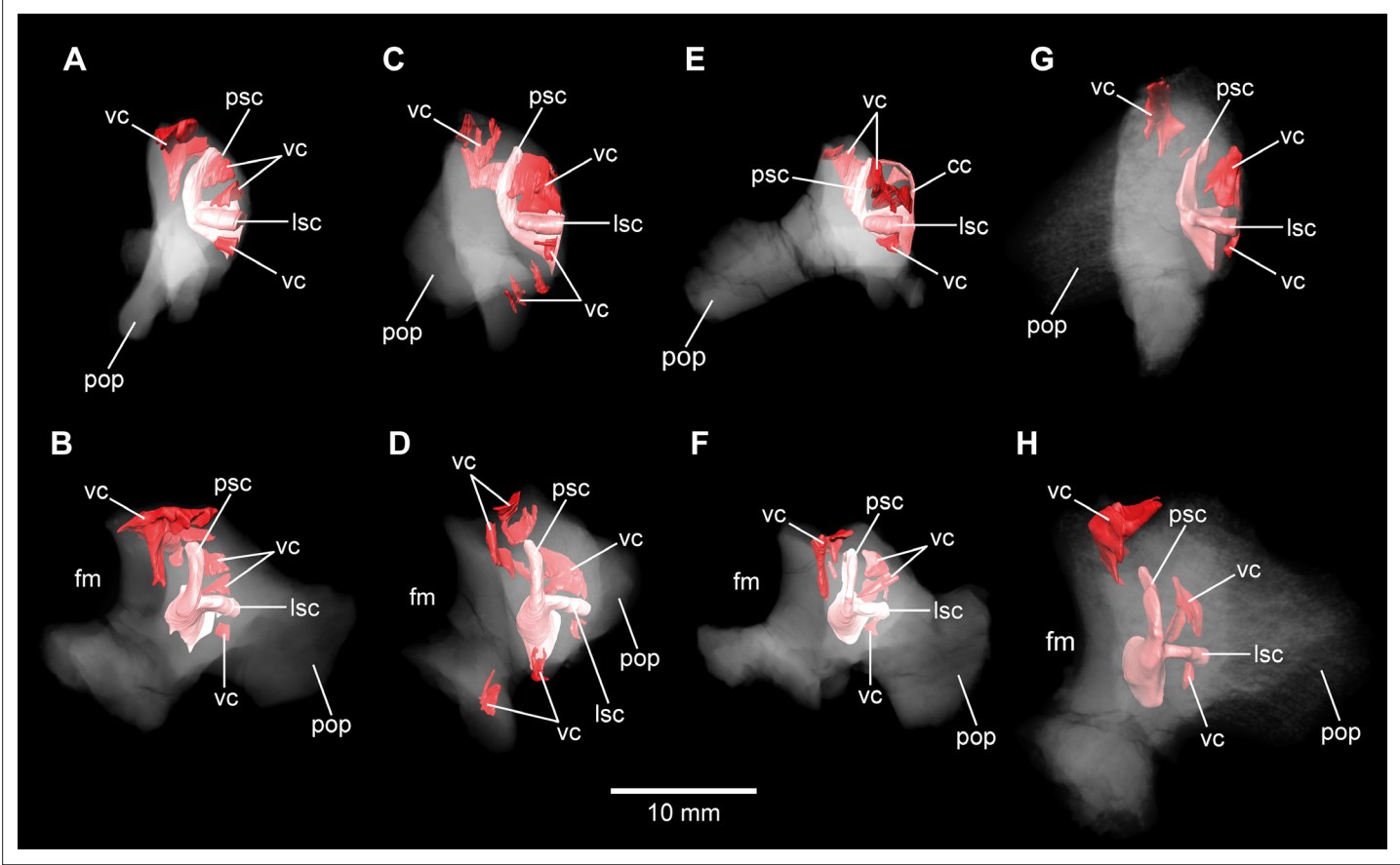

**Figure 8.** *Europasaurus holgeri*, 3D models of the posterior portions of the endosseous labyrinth in (**A,B**) DFMMh/FV 898, (**C,D**) DFMMh/FV 981.2, (**E,F**) DFMMh/FV 249, and (**G,H**) DFMMh/FV 205 in (**A,C,E,G**) anterolateral and (**B,D,F,H**) posterior view. Note that scale mainly applies to posterior perspective (**B,D,F,H**). fm, foramen magnum; lsc, lateral semicircular canal; pop, paroccipital process; psc, posterior semicircular canal; vc, vascular cavity.

The online version of this article includes the following figure supplement(s) for figure 8:

**Figure supplement 1.** *Europasaurus holgeri*, isolated otoccipitals (DFMMh/FV 898, **A,B**; DFMMh/FV 981.2, **C,D**; DFMMh/FV 249, **E,F**; DFMMh/FV 205, **G,H**) in (**A,C,E,G**) posterior and (**B,D,F,H**) anterior view.

**Figure supplement 2.** *Europasaurus holgeri*, 3D models of isolated otoccipitals and inner features (DFMMh/FV 898, **A**; DFMMh/FV 981.2, **B,C**; DFMMh/FV 249, **D**; DFMMh/FV 205, **E**) in (**A,B,D,E**) anterodorsomedial and (**C**) ventral view.

in DFMMh/FV 581.1. The ventral aspect of the endocast is anterodorsally inclined and bears a median ridge (becoming mediolaterally narrower in anterior direction; *Figures 1A, 2C,3A*), reaching between the foramen magnum and the anteroventral portion of the endocast (not considering the pituitary). Posteroventral to the abducens nerve (CN VI), a single median protuberance is present on the endocast, produced by a fossa on the floor of the endocranial cavity (*Figures 2C and 3*; *Figure 1—figure supplements 2 and 3*). In addition, anterodorsally to the proximal openings for the abducens nerve (CN VI), a single median opening is present on the braincase floor, producing a connection to the pituitary fossa (probably for vascularization; see *Paulina-Carabajal, 2012*; *Sues et al., 2015* for arguments on arterial or venous identity). The general osteological configuration of the endocranial floor (*Figure 1—figure supplements 2 and 3*) seems very similar in the macronarian *Giraffatitan* (*Janensch, 1935*: Figure 117). The anterodorsally incomplete endocranial cavity of DFMMh/FV 581.1, 2, and 3 comprises a volume of about 35 cm³ (including the pituitary fossa). On the ventral aspect of DFMMh/FV 581.1, a small funnel-like depression anterior to the occipital condyle ends blindly (*Figure 2D*).

## Endosseous labyrinth
Both vestibular systems are preserved and are ventrally connected to the respective cochlea in DFMMh/FV 581.1 (the semicircular canals of the left inner ear were only vaguely perceptible in some

places). Whereas only the left endosseous labyrinth is preserved in DFMMh/FV 1077, only the right one is preserved within DFMMh/FV 466+205. The following description is based on the mentioned endosseous labyrinths (*Figure 6*). The vertical semicircular canals are relatively long and slender. Dorso-ventrally, the ASC reaches considerably higher than the posterior one, and the ASC occupies more of the anteroposterior length of the vestibular system. The common crus is dorsally slightly posteriorly inclined (where preserved). While the posterior semicircular canal (PSC) forms a low arc, the ASC turns about 180° to contact the common crus dorsomedially. The medial aspect of the common crus is exposed to the endocranial cavity in DFMMh/FV 581.1 and DFMMh/FV 1077 (*Figures 4 and 6G*; *Figure 1—figure supplement 4*; *Figure 4—figure supplement 2*). The angle between the ASC and the PSC amounts 80° (measured in dorsal view with the common crus as fixpoint). The LSC is antero-posteriorly short. In dorsal view, its anterior ampulla appears posteriorly shifted, producing a medi-ally concave gap between the ASC and LSC (*Figure 6B,F,J*). Such a medial concavity is also present between the LSC and the PSC (best seen in dorsal view). The cochlear duct is approximately as high as the vestibular system dorsoventrally, points anteroventrally and very slightly medially (in DFMMh/FV 581.1 and 1077). In lateral view, the cochlear duct is anteroposteriorly slender with sub-parallel anterior and posterior margins. However, mediolaterally, the cochlear duct is very wide, resulting in an elongated oval-shaped cross-section. The fenestra ovalis (*Figures 1A, B, 2A, B and 6*; *Figure 4—figure supplement 1*) is situated close to the dorsoventral mid-length of the lateral aspect of the cochlear duct (in DFMMh/FV 581.1 and DFMMh/FV 1077). This is also true for the anteroposteriorly oriented fenestra pseudorotunda (*Figures 4 and 6D*; *Figure 1—figure supplement 4*), lying on the posteromedial aspect of the cochlear duct. The hiatus acusticus expresses as an anteromedially open notch (similar to the theropod *Irritator*; *Schade et al., 2020*) on the actual fenestra pseudorotunda in DFMMh/FV 581.1 (*Figure 1—figure supplement 4*).

## Auditory capabilities

To get a rough idea of the audition of *Europasaurus*, we measured the dorsoventral cochlear duct length of DFMMh/FV 581.1 (c. 16 mm; as outlined by *Walsh et al., 2009*; however, see *Gleich et al., 2005*; *Witmer and Ridgely, 2008a*; *Paulina-Carabajal et al., 2016*) and the anteroposterior basi-cranial length (c. 55 mm; from the anterodorsal part of the pituitary fossa to the posterior-most part of the occipital condyle). Based on the equations of *Walsh et al., 2009*, our estimate of the mean hearing frequency of *Europasaurus* yields a value of 2225 Hz and a frequency bandwidth of 3702 Hz (374–4076 Hz). The auditory capabilities of the Late Triassic early-diverging sauropodomorph *Theco-dontosaurus* from England was estimated by same means with a mean frequency of 1893 Hz and a band width of 3089 Hz (349–3438 Hz; *Ballell et al., 2021*).

## Inner ears and cavities of incomplete specimens

In addition to DFMMh/FV 581.1, 2, and 3 (*Figures 1–3* and *Figure 6A–D*; *Figure 1—figure supplements 1–4*), eight other braincase specimens (that hold parts of the endosseous labyrinth), assigned to *Europasaurus*, were scanned and analysed. DFMMh/FV 1077 (*Figures 4 and 6E–H*; *Figure 4—figure supplements 1 and 2*) contains a complete left endosseous labyrinth and was categorized as belonging to an osteological mature individual in *Marpmann et al., 2014*; as DFMMh/FV 581.1, 2, and 3. Furthermore, there are two right elements (*Figures 5, 6I–L, 7A,B, 8G,H*; *Figure 5—figure supplement 1*; *Figure 7—figure supplements 1A, B and 2A*; *Figure 8—figure supplements 1G, H and 2E*; DFMMh/FV 205, a fragmentary otoccipital, and DFMMh/FV 466, a fragmentary prootic) that were originally found some 10 cm apart from each other in the sedimentary matrix. Whereas DFMMh/FV 205 was thought to belong to a juvenile, DFMMh/FV 466 was supposed to belong to a consid-erably older individual (both estimations are mainly based on size and surface texture; *Marpmann et al., 2014*). However, DFMMh/FV 205 and DFMMh/FV 466 articulate well with each other and jointly contain most of the endosseous labyrinth and the dorsal portion of the lagena, all in a meaningful manner in respect to size, position, and orientation of its compartments. DFMMh/FV 466 is of similar size and texture as the other prootics considered here. DFMMh/FV 205 is considerably smaller than the otoccipitals in the adult specimens. Hence, DFMMh/FV 466+205 are herein interpreted to belong to the same juvenile individual. Furthermore, there are two left fragmentary prootics (*Figure 7C–F*; *Figure 7—figure supplement 1C–F*, *Figure 7—figure supplement 2B, C*; DFMMh/FV 561 and DFMMh/FV 964) containing most of the ASC, the ventral base of the common crus, the anterior

ampulla of the LSC, and the anterior base of the lagena; both specimens were assigned to relatively mature individuals (*Marpmann et al., 2014*). The three right fragmentary otoccipitals DFMMh/FV 249, DFMMh/FV 898, and DFMMh/FV 981.2 (*Figure 8A–F*; *Figure 8—figure supplements 1A–F and 2A–D*) contain at least the posterior parts of the LSC and the lagena, as well as most of their PSCs; these specimens were assigned to immature individuals (*Marpmann et al., 2014*).

In general, the morphology of the inner ears contained within these isolated specimens is consistent to what can be observed in DFMMh/FV 581.1 and DFMMh/FV 1077. Since *Marpmann et al., 2014*, used the vascularization (indicated by surface texture) of *Europasaurus* specimens as a critical character in judging the relative maturity, the inner cavities surrounding the endosseous labyrinths were examined herein.

No discrete cavities could be found in DFMMh/FV 581.1, DFMMh/FV 1077, DFMMh/FV 964, and DFMMh/FV 561 (all considered to represent more or less mature individuals). The otoccipitals DFMMh/FV 249, DFMMh/FV 898, and DFMMh/FV 981.2 and the articulated specimens DFMMh/FV 205 (otoccipital) and DFMMh/FV 466 (prootic) show very similar, or corresponding, patterns of inner cavities (*Figures 7 and 8*; *Figure 7—figure supplement 2*; *Figure 8—figure supplement 2*). All four otoccipitals show dorsoventrally deep cavities posterodorsal to anteromedial to the PSC, close to the articulation surface with the supraoccipital (except for DFMMh/FV 205, in which this cavity network is not as much extended anteriorly). There are T- (DFMMh/FV 898 and DFMMh/FV 981.2), V- (DFMMh/FV 205), or X- (DFMMh/FV 249) shaped (in cross-section in anterior view), dorsoventrally high and mediolaterally thin structures anterior to the PSC and dorsal to the LSC (close to the articulation surface with the prootic). Additionally, all four otoccipital specimens bear relatively small cavities ventral to the LSC (again, close to the articulation surface with the prootic). DFMMh/FV 981.2 shows dorsoventrally high cavities posteroventrally to the endosseous labyrinth (close to the articulation surface with the basioccipital). Generally, the cavities, likely of vascular purpose, of DFMMh/FV 205 seem not as large and extensive as in the other three otoccipitals. This coincides with their size and assumed relative maturity (DFMMh/FV 205 being the largest, smoothest and, hence, most mature of them; see *Marpmann et al., 2014*). Whereas DFMMh/FV 466 bears a small cavity ventral to the LSC (corresponding to the respective cavity in the otoccipital DFMMh/FV 205), no other unequivocal cavities could be found, which is surprising when the V-shaped cavity close to the prootic contact of DFMMh/FV 205 is considered.

## Discussion

### Comparison of neurovascular anatomy and potential ecological implications

Although not as prominent as in *Dicraeosaurus* (*Janensch, 1935*; *Paulina Carabajal et al., 2018*) and some specimens of *Diplodocus* (*Witmer and Ridgely, 2008a*), the position and morphology of the dorsal expansion of *Europasaurus* gives a rather 'upright' or sigmoidal appearance to the endocast (*Figure 1A*; see also *Paulina-Carabajal et al., 2020*). This is partly explained by the (preservational) lack of its olfactory bulb and tract. The first cranial nerve is not expected to be very long in many sauropods, especially in the closely related macronarian taxa *Camarasaurus* and *Giraffatitan* (*Witmer and Ridgely, 2008a*; *Knoll and Schwarz-Wings, 2009*; see also *Müller, 2021*). In contrast, the braincase endocast is rather tubular in some taxa, for example, the early-diverging sauropodomorph *Buriolestes* (*Müller et al., 2021*), the rebbachisaurid *Nigersaurus* (*Sereno et al., 2007*), and the titanosaur specimen MCCM-HUE-1667 (*Knoll et al., 2015*). Instead, the endocast of *Europasaurus* seems to be most similar to *Giraffatitan* (*Janensch, 1935*; *Knoll and Schwarz-Wings, 2009*; formerly *Brachiosaurus brancai*, see *Paul, 1988*; *Taylor, 2009*).

Contrary to other sauropod taxa (e.g., *Spinophorosaurus*, *Diplodocus*, *Camarasaurus*, and *Sarmientosaurus*; see *Witmer et al., 2008b*; *Knoll et al., 2012*; *Martínez et al., 2016*), there are no discrete canals for vascular features such as, for example, the rostral middle cerebral vein or the orbitocerebral vein on the endocast of *Europasaurus*.

A ventral ridge on the medulla, as seen in *Europasaurus* (*Figures 1A, 2C, 3A*), seems to be present, although not as pronounced, in *Thecodontosaurus* (*Ballell et al., 2021*), the early-diverging sauropod specimen OUMNH J13596 (*Bronzati et al., 2018*), *Spinophorosaurus* (*Knoll et al., 2012*),

*Camarasaurus* (*Witmer and Ridgely, 2008a*) and, potentially, *Giraffatitan* (*Janensch, 1935*; *Knoll and Schwarz-Wings, 2009*) as well.

Although not obvious on the endocast (*Knoll and Schwarz-Wings, 2009*), *Giraffatitan* seems to bear a median fossa posteromedially to the proximal CN VI openings (*Janensch, 1935*: Figure 117); the respective protuberance in *Europasaurus* marks a distinct kink on the endocast (*Figures 2C and 3*; *Figure 1—figure supplements 2 and 3*).

The endocast of *Europasaurus* bears two pairs of canals on the ventrolateral aspect of the pituitary, the posterior of which is interpreted to represent the internal carotid here (*Figures 1A, 2,3*; *Figure 1—figure supplement 1*; in accordance with *Marpmann et al., 2014*: Figure 13A). Whereas structures identified as the craniopharyngeal canal are present anterior to the carotid artery in the titanosaur specimen CCMGE 628/12457 (*Sues et al., 2015*) and the diplodocid specimen MMCh-Pv-232 (assigned to *Leinkupal*; *Garderes et al., 2022*), they are situated posteriorly in *Apatosaurus* (*Balanoff et al., 2010*; see also *Paulina-Carabajal, 2012*, and *Paulina Carabajal et al., 2014* for subcondylar foramina in the vicinity of the internal carotid arteries). However, in these taxa, the craniopharyngeal canal is a singular median canal. This may render the anterior of the two pairs of canals on the ventral aspect of the pituitary in *Europasaurus* the canals for the neurohypophysis (*Paulina-Carabajal, 2012*). The pituitary of the *Europasaurus* endocast of DFMMh/FV 581.1 does not project much more ventrally than the posteroventral margin of the medulla oblongata. The pituitary is slightly higher dorsoventrally than the ASC (*Figure 1A*). Usually in sauropods, the pituitary is large and inclined posteroventrally, reaching much more ventrally than the ventral margin of the hindbrain (see, e.g., *Knoll and Schwarz-Wings, 2009*; *Martínez et al., 2016*; see also *Sues et al., 2015* for an extreme reached in the titanosaur specimen CCMGE 628/12457 with a short ASC and an enormous pituitary). The finding of a relatively small pituitary fossa in *Europasaurus* and early-diverging sauropodomorphs seem to support a close connection of body and pituitary size, as suggested by some authors (*Nopcsa, 1917*; *Edinger, 1942*; *Müller et al., 2021*). The microCT data of DFMMh/FV 581.1 suggest that the right CN VI canal closely passes by the pituitary fossa without a penetration, whereas the left CN VI canal tangents on the pituitary fossa and opens into the latter (*Figure 1—figure supplement 3*). The feature of the CN VI canals not penetrating the pituitary fossa seems typical for titanosaurs (e.g., *Paulina-Carabajal, 2012*; *Knoll et al., 2015*; *Paulina-Carabajal et al., 2020*). Whereas *Knoll and Schwarz-Wings, 2009* note such a penetration or connection on the endocast MB.R.1919, *Janensch, 1935*, originally described penetrating canals in the *Giraffatitan* braincase specimens S 66 (on which the endocast MB.R.1919 is based) and Y 1. However, the *Giraffatitan* braincase specimen t 1 seems to show CN VI canals rather passing by the pituitary fossa (Janensch, 1935). This may suggest a certain role of individual expressions (*Giraffatitan*), asymmetries (*Europasaurus*), and/or represents a phylogenetically potentially reasonable intermediate state (nonetheless, this feature may also be prone to preservational bias).

The endosseous labyrinth of *Europasaurus* (*Figure 6*) is most similar to *Giraffatitan* (*Janensch, 1935*) and *Spinophorosaurus* (*Knoll et al., 2012*) in bearing a relatively long ASC and a long lagena. In dorsal view, the anterior ampulla of the short LSC in *Europasaurus* displays a medially concave gap between the ASC and LSC (*Figure 6B,F,J*). Similarly, a pronounced concavity is present between the LSC and the PSC (best seen in dorsal view). Both concave gaps (the ASC and the PSC project further laterally than the lateral outline of the LSC reaches medially) are similarly present in many Titanosauriformes (with the exception of FAM 03.064; *Knoll et al., 2019*): *Giraffatitan* (*Janensch, 1935*), *Malawisaurus* (*Andrzejewski et al., 2019*), *Sarmientosaurus* (*Martínez et al., 2016*), CCMGE 628/12457 (*Sues et al., 2015*), *Jainosaurus* (*Andrzejewski et al., 2019*), *Ampelosaurus* (*Knoll et al., 2013*), *Narambuenatitan* (*Paulina-Carabajal et al., 2020*), *Bonatitan*, *Antarctosaurus*, MCF-PVPH 765 and MGPIFD-GR 118 (*Paulina-Carabajal, 2012*), but also in the rebbachisaurids *Limaysaurus* and *Nigersaurus* (*Paulina-Carabajal and Calvo, 2021*). In contrast to other sauropods, the anterior portion of the LSC, as well as its lateral-most extent (best seen in dorsal view), seems somewhat posteriorly shifted in the macronarians *Camarasaurus* (*Witmer and Ridgely, 2008a*) and *Europasaurus* (*Figure 6B,F,J*; for further discussion, see *Supplementary file 1*).

Although the mediolateral width of the lagena does not appear to be associated with auditory capabilities (*Walsh et al., 2009*), the lagena of *Europasaurus* is conspicuously thick mediolaterally, especially when compared to its anteroposterior slenderness (*Figure 6*). The calculated auditory capacities (based on *Walsh et al., 2009*) impute *Europasaurus* a relatively wide hearing range with

a high upper frequency limit (among non-avian dinosaurs; *Lautenschlager et al., 2012*; *King et al., 2020*; *Sakagami and Kawabe, 2020*). *Walsh et al., 2009*, demonstrate a certain correlation between hearing range, complexity of vocalization, and aggregational behaviour in extant reptiles and birds (see also *Gleich et al., 2005*; *Hanson et al., 2021*). Following their conclusions and other studies suggesting (age-segregated) gregariousness in sauropodomorphs on the basis of nesting sites, body, and ichnofossils (e.g., *Lockley et al., 2002*; *Sander et al., 2008*; *Myers and Fiorillo, 2009*; *Pol et al., 2021a*), it appears plausible that *Europasaurus* lived in groups with conspecifics (although it is not clear whether this took place perennial or seasonal, e.g., for 'brooding'), which made airborne communication crucial. Furthermore, taphonomic reasons (femora count suggests at least 21 individuals in close temporal and spatial connection with very young and very old individuals being rarely represented; *Scheil et al., 2018*) and evidence for two morphotypes in the cranial and postcranial material of *Europasaurus* may suggest some form of social cohesion (*Carballido and Sander, 2013*; *Marpmann et al., 2014*). However, while a given species is likely to perceive sounds within the frequency spectrum it is able to produce, it may be rather unlikely that the full range of frequencies that can be heard is covered by the sound production ability (see also *Walsh et al., 2009*; *Senter, 2008*). Habitat preferences potentially play a role as well: 'acoustically cluttered' habitats like forests seem associated with a tendency towards high-frequency intraspecific communication in recent mammals (*Charlton et al., 2019*). Together with tropic Late Jurassic conditions in Europe (*Armstrong et al., 2016*), this may be part of the explanation of the recovered auditory capacities of *Europasaurus*.

## Fragmentary bones and their eco-ontogenetic meaning

An interesting issue are the different morphological ontogenetic stages of DFMMh/FV 466 and DFMMh/FV 205 mentioned in *Marpmann et al., 2014*. The authors considered the prootic DFMMh/FV 466 more mature than the otoccipital DFMMh/FV 205. Indeed, DFMMh/FV 466 is about as large as the prootics of DFMMh/FV 581.1, DFMMh/FV 1077, DFMMh/FV 964, and DFMMh/FV 561 (*Figure 7*; *Figure 7—figure supplement 1*), but the otoccipital DFMMh/FV 205 is much smaller than the ones in DFMMh/FV 581.1 and DFMMh/FV 1077 (and only slightly larger than DFMMh/FV 981.2, DFMMh/FV 898, and DFMMh/FV 249; *Figure 8*; *Figure 8—figure supplement 1*).

In addition to general size of the specimens, and build and rugosity of articular facets, *Marpmann et al., 2014* (see also *Benton et al., 2010*) defined the morphological ontogenetic stages also by bone surface smoothness, advocating for vascularization: the smoother the surface, the lesser the degree of vascularization and – in tendency – the more mature the individual bone. Our findings support this (*Figures 7 and 8*; *Figure 7—figure supplement 2*; *Figure 8—figure supplement 2*). While the bases of individual cavities described herein may represent depressions of articulation areas, their deep penetration into the bone is unambiguous. Apart from this, the described structures might represent sutures. However, the position and orientation of individual cavities do not conform to what would be expected. Since these cavities make sense in the frame of morphological ontogenetic stages used in *Marpmann et al., 2014*, they are considered as so far unknown vascular expressions of juvenile *Europasaurus* individuals here.

DFMMh/FV 466 and DFMMh/FV 205 articulate very well with each other, especially on their lateral aspects. Additionally, there are cavities ventral to the LSC that seem to have been continuous originally (*Figure 5*; *Figure 7A, B*; *Figure 8G, H*, *Figure 5—figure supplement 1*). However, whereas the fenestra ovalis is considerably smaller than the vagal foramen in DFMMh/FV 581.1 and DFMMh/FV 1077 (*Figure 3B*; *Figure 4—figure supplement 1A*), it seems that in DFMMh/FV 466+205 this is vice versa (although this impression may be due to the fragmentary nature of the latter two specimens; *Figure 5L*; *Figure 5—figure supplement 1F*). If DFMMh/FV 466 and DFMMh/FV 205 are in articulation, there is a large gap on their common dorsal aspect (*Figure 5J*; *Figure 5—figure supplement 1H*). Considering DFMMh/FV 581.1 and DFMMh/FV 1077 and, for example, the braincase of *Massospondylus* (*Chapelle and Choiniere, 2018*), the supraoccipital usually occupies this gap. In case our interpretation of a common bond between DFMMh/FV 466 and DFMMh/FV 205 is misleading and they actually represent two differently matured individuals, it is still noticeable that the preserved parts of the conjoined endosseous labyrinth of DFMMh/FV 466 and DFMMh/FV 205 displays the same general features as DFMMh/FV 581.1 and DFMMh/FV 1077 and is anteroposteriorly almost as long as the latter two specimens (*Figures 5 and 6*; *Supplementary file 1*). This suggests an allometric growth between the prootic and otoccipital: during growth, the prootic reaches the 'adult' size faster

than the otoccipital, producing a surprisingly small paroccipital process (or a surprisingly large prootic) in juvenile individuals of *Europasaurus* (seemingly, also seen in *Massospondylus*; *Sues et al., 2004*), containing a relatively large endosseous labyrinth (see also *Fabbri et al., 2021*, for ontogenetic transformations in the cranium of sauropodomorphs). A relatively large immature endosseous labyrinth seems also to be present in the ornithischians *Dysalotosaurus* (*Lautenschlager and Hübner, 2013*), *Psitaccosaurus* (*Bullar et al., 2019*), and *Triceratops* (*Morhardt et al., 2018*). Furthermore, the endosseous labyrinth appears relatively large in juveniles of *Massospondylus* (*Neenan et al., 2019*) and the extant, precocial ostrich (*Romick, 2013*), and stays morphologically relatively stable throughout ontogeny (see also *Jeffery and Spoor, 2004*).

The vestibular apparatus detects movements with the aid of endolymphatic fluid and cilia contained within the semicircular canals, which is crucial for locomotion (see, e.g., *Benson et al., 2017*). Thus, a relatively large and morphologically adult-like endosseous labyrinth in expectedly very young individuals of *Europasaurus* suggests that hatchlings had to be light on their feet very fast in this dwarfed sauropod taxon.

## Conclusion

*Europasaurus* has a rather sigmoid general braincase endocast shape, with a comparably large dorsal expansion, two openings for CN XII, an angle of 50° between the pituitary fossa and the LSC, and the ASC is clearly dorsoventrally higher than the PSC (*Figures 1–4* and *Figure 6*). This and additional novel details, such as the highly uniform vascular cavities within the juvenile braincase material (*Figures 7 and 8*; *Figure 7—figure supplement 2*; *Figure 8—figure supplement 2*), add to our knowledge about dinosaur neuroanatomy. The relatively small pituitary fossa (*Figure 1A*) in an insular dwarf lends support to the old idea of being a proxy for body size (*Nopcsa, 1917*; *Edinger, 1942*; *Müller et al., 2021*).

Many sauropods were extremely large land-dwellers as adults, and still, started as tiny hatchlings, indicating enormously fast growth rates (e.g., *Carpenter, 1999*; *Hallett and Wedel, 2016*; *Curry Rogers et al., 2016*). The threat arising from the discrepancy of several tens of tons between adults and juveniles makes it, among other reasons, unlikely that these animals were able to take good care for their offspring (e.g., *Sander et al., 2011*; *Curry Rogers et al., 2016*). This implies a great mobility early in life (precociality in a broader sense; see *Dial, 2003*; *Iwaniuk and Nelson, 2003*) of sauropods (*Sander et al., 2011*). Although *Europasaurus* represents an island dwarf (adults were probably not as dangerous for their juveniles), having roamed islands not exceeding an area of three times modern-day Bavaria (*Sander et al., 2006*), this taxon seemingly retained characteristics potentially associated with precociality (and therefore potentially r-strategy; *Sander et al., 2008*; *Myers and Fiorillo, 2009*; *Hallett and Wedel, 2016*) from its large-bodied ancestors. As also suggested by the taphonomic circumstances (*Sander et al., 2006*; *Carballido and Sander, 2013*; *Marpmann et al., 2014*; *Scheil et al., 2018*; see also *Supplementary file 1*), *Europasaurus* individuals likely stayed in a certain social cohesion, and potentially practiced colonial nesting as is known from other sauropodomorphs (*Lockley et al., 2002*; *Sander et al., 2008*; *Myers and Fiorillo, 2009*; *Pol et al., 2021a*). In concert with the approximate auditory capabilities offered here, our findings add hints towards the nature of aggregation with a certain complexity of reproductive and social behaviours for these little real-life titans, thriving in Europe some 154 Ma ago.

## Materials and methods

The articulated braincase specimen of *E. holgeri*, DFMMh/FV 581.1, together with both loose parietals (DFMMh/FV 581.2 and 3), is traversed by breakages but not strongly deformed, lacking parts of the anterior and dorsomedial skull roof, as well as the anteromedial walls of the endocranial cavity. The articulated and assembled braincase lacks the frontals, the right orbitosphenoid and laterosphenoid. The parietals are anteriorly, posterodorsomedially, and posteriorly incomplete and somewhat deformed (if they fit posteromedially with the supraoccipital they do not fit with the supraoccipital, prootic and laterosphenoid further anteriorly, and vice versa). The braincase of, for example, the macronarian *G. brancai* suggests a plain posterior skull roof not exceeding the dorsal extent of the sagittal nuchal crest; this served as an orientation here.

## Macro-photography

All specimens, except DFMMh/FV 581.1, 2, and 3, were documented using a Canon EOS 70D reflex camera equipped with a Canon EFS 10–135 mm objective, extension tubes (13 or 21 mm), and a Canon Macro Twin Lite MT-26EX-RT. Light was cross-polarized in order to reduce reflections of the specimen surface. Images were recorded in different focal planes (z-stacks) and subsequently fused with CombineZP (Alan Hadley). All obtained images were optimized for colour balance, saturation, and sharpness using Adobe Photoshop CS2.

## Micro-computed tomography

MicroCT of DFMMh/FV 581.1, 2, and 3 (*Figures 1–3*; *Figure 1—figure supplements 1–4*) was performed using a Metrotom 1500 (Carl Zeiss Microscopy GmbH, Jena, Germany) in a subsidiary of Zeiss in Essingen; 1804 images were recorded with binning 1 resulting in a DICOM data set (for further details of settings and voxel size, see *Supplementary file 1*).

All other specimens (*Figures 4, 5, 7 and 8*; *Figure 4—figure supplements 1 and 2*; *Figure 5— figure supplement 1*, *Figure 7—figure supplements 1 and 2*; *Figure 8—figure supplements 1 and 2*) were documented with a Xradia MicroXCT-200 (Carl Zeiss Microscopy GmbH, Jena, Germany) of the Imaging Center of the Department of Biology, University of Greifswald; 1600 projection images were recorded each, using 0.39× objective lens, with binning 2 (for further details of settings and voxel size for each specimen, see *Supplementary file 1*). The tomographic images were reconstructed with XMReconstructor software (Carl Zeiss Microscopy GmbH, Jena, Germany), binning 1 (full resolution) resulting in image stacks (TIFF format).

Digital segmentation and measurements were produced utilizing the software Amira (5.6), based on DICOM files (DFMMh/FV 581.1, 2, and 3) and tiff files (remaining material). The microCT data were manually segmented to create 3D surface models. In DFMMh/FV 581.1, 2, and 3, the X-ray absorption of the fossil and the sediment within is quite similar, resulting in low contrast in many places. Furthermore, for preservational reasons (lack of both frontals, right orbitosphenoid, laterosphenoid, and loose parietals), the extent of the digital model of the endocast was conservatively estimated on the skull roof and on the anterodorsal region; some asymmetries on the endocast are explained by this circumstance.

## Acknowledgements

We are extremely thankful towards Zeiss in Essingen (especially Bastian Zwick and Stephan Tomaschko), the Universitätsmedizin in Greifswald (especially Christopher Nell) for actuating their CT devices for the fossils of *Europasaurus*; further microCT were performed at the Imaging Center of the Department of Biology, University of Greifswald (DFG INST 292/119-1 FUGG; DFG INST 292/120-1 FUGG). We thank Michael 'Ede' Kenzler, Jakob Krieger, Georg Brenneis, Jennifer Legat, Steffen Harzsch, and Ingelore Hinz-Schallreuter (all University of Greifswald, Germany), together with Benjamin Englich (Dinosaurierpark Münchehagen, Germany) and Serjoscha Evers (University of Fribourg) for their support and discussions. Additionally, we are grateful for the editor's and referee's valuable contributions.

## Additional information

### Competing interests

Nils Knötschke: Nils Knötschke is affiliated with Mineralientage. The author has no financial interests to declare. The other authors declare that no competing interests exist.

### Funding

| Funder | Grant reference number | Author |
|---|---|---|
| Universität Greifswald | Bogislaw scholarship | Marco Schade |
| Deutsche Forschungsgemeinschaft | DFG INST 292/119-1 FUGG | Marie K Hörnig |

| Funder | Grant reference number | Author |
|---|---|---|
| Deutsche Forschungsgemeinschaft | DFG INST 292/120-1 FUGG | Marie K Hörnig |

The funders had no role in study design, data collection and interpretation, or the decision to submit the work for publication.

## Author contributions

Marco Schade, Conceptualization, Resources, Data curation, Software, Formal analysis, Investigation, Visualization, Methodology, Writing – original draft, Project administration, Writing – review and editing; Nils Knötschke, Supervision, Validation, Investigation, Writing – original draft; Marie K Hörnig, Resources, Software, Investigation, Visualization, Methodology, Writing – original draft; Carina Paetzel, Software, Formal analysis, Investigation, Visualization, Methodology, Writing – original draft; Sebastian Stumpf, Software, Formal analysis, Investigation, Visualization, Methodology, Writing – original draft, prepared the figures and contributed to the manuscript

## Author ORCIDs

Marco Schade http://orcid.org/0000-0003-1658-6854
Sebastian Stumpf http://orcid.org/0000-0002-1945-2387

## Decision letter and Author response

Decision letter https://doi.org/10.7554/eLife.82190.sa1
Author response https://doi.org/10.7554/eLife.82190.sa2

## Additional files

### Supplementary files

- Supplementary file 1. Supplementary information, scan details, and measurements.
- MDAR checklist

### Data availability

The microCT data and neuroanatomical models of all fossil specimens depicted herein are published online, in the repository MorphoSource (Europasaurus holgeri - neuroanatomy - DFMMh/FV - Schade et al. 2023 // MorphoSource).

The following dataset was generated:

| Author(s) | Year | Dataset title | Dataset URL | Database and Identifier |
|---|---|---|---|---|
| Marco S | 2022 | Europasaurus holgeri - neuroanatomy - DFMMh/FV - Schade et al. 2023 | https://www.morphosource.org/projects/000445173?locale=en | morphosource, 000445173 |

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
