## [Editor Report]

The authors provide the first detailed description of the neuroanatomy of the remarkable dwarf sauropod *Europasaurus* from the Jurassic of Germany, which, at least in this regard, was not very different from some of its much larger relatives. The available evidence is compelling and convincing. The comparative sections of the manuscript are solid and provide a relatively broad overview. Based on remains of different individuals and growth stages, the authors suggest that *Europasaurus* was likely precocial. The authors also assess the likely auditory capabilities and their relevance to the reproductive and social behaviour of this island-dwelling dinosaur.

---

## [Decision Letter]

**Decision letter after peer review:**

Thank you for submitting your article "They call me the wanderer – Neurovascular anatomy of dwarfed dinosaur implies precociality in sauropods" for consideration by *eLife*. Your article has been reviewed by 2 peer reviewers, and the evaluation has been overseen by a Reviewing Editor and George Perry as the Senior Editor. The reviewers have opted to remain anonymous.

The reviews are largely positive and this work is of broad interest. Addressing essential revisions should not be too difficult but do not hesitate to reach out if you have any additional questions.

Essential revisions:

1) Provide a more robust comparative overview of endocast/semicircular canal changes across several representative sauropods (see reviewer comments below).

2) Provide more information on limitations of certain interpretations (e.g., regarding gregarious behaviour, and hearing frequencies see also below), or provide additional information to address some of the reviewer comments.

*Reviewer #1 (Recommendations for the authors):*

Line 90-91: "see20 for correlation of brain and endocast". There are several manuscripts that precede the reference listed by the authors dealing with this issue and confirming that, although a perfect fitting is not present, the morphology of the endocast in non-avian dinosaurs is still a pretty good match. I would ask the authors to cite the following studies:

– Evans, D. C. (2005). New evidence on brain-endocranial cavity relationships in ornithischian dinosaurs. Acta Palaeontologica Polonica, 50(3).

– Witmer, L. M., and Ridgely, R. C. (2008). Structure of the brain cavity and inner ear of the centrosaurine ceratopsid dinosaur Pachyrhinosaurus based on CT scanning and 3D visualization.

– Morhardt, A. C. (2016). Gross Anatomical Brain Region Approximation (GABRA): Assessing Brain Size, Structure, and Evolution in Extinct Archosaurs (Doctoral dissertation, Ohio University).

– Fabbri, M., Mongiardino Koch, N., Pritchard, A. C., Hanson, M., Hoffman, E., Bever, G. S., … and Bhullar, B. A. S. (2017). The skull roof tracks the brain during the evolution and development of reptiles including birds. Nature ecology and evolution, 1(10), 1543-1550.

Line 200-205: would it be possible to have a table of comparisons between hearing frequencies between Europasaurus and other sauropodomorphs? This would give a better idea of the peculiarity of Europasaurus in comparison to other sauropodomorphs and will foster a better understanding of the ecology of these animals.

Line 320-333: although I agree that modern species are the key to inferring ecological behavior in extinct ones, I would ask the authors to discuss additional scenarios. For example, what if this taxon was simply communicating during the breeding season and was solitary for the majority of its life? Precociality includes this kind of behavior, but this is not discussed at the moment. I would suggest the authors expand this discussion

Figures: these are generally well done. That being said, I would strongly suggest a final figure in the main text showing multiple endocasts and semicircular canals of multiple sauropodomorphs in a phylogenetic context. This will help the general reader to understand the evolutionary trends affecting the evolution of the brain in this clade. I would be happy even with schematic drawings showing multiple taxa.

*Reviewer #2 (Recommendations for the authors):*

Overall, the manuscript is well-presented and of high quality. The detailed descriptions are particularly appreciated by this reviewer. The results indeed give important insight into the inspected aspects and the study represents a very important step forward in the knowledge of sauropod neuroanatomy and Europasaurs.

---

## [Author Response]

Reviewer #1 (Recommendations for the authors):Line 90-91: "see20 for correlation of brain and endocast". There are several manuscripts that precede the reference listed by the authors dealing with this issue and confirming that, although a perfect fitting is not present, the morphology of the endocast in non-avian dinosaurs is still a pretty good match. I would ask the authors to cite the following studies:– Evans, D. C. (2005). New evidence on brain-endocranial cavity relationships in ornithischian dinosaurs. Acta Palaeontologica Polonica, 50(3).– Witmer, L. M., and Ridgely, R. C. (2008). Structure of the brain cavity and inner ear of the centrosaurine ceratopsid dinosaur Pachyrhinosaurus based on CT scanning and 3D visualization.– Morhardt, A. C. (2016). Gross Anatomical Brain Region Approximation (GABRA): Assessing Brain Size, Structure, and Evolution in Extinct Archosaurs (Doctoral dissertation, Ohio University).– Fabbri, M., Mongiardino Koch, N., Pritchard, A. C., Hanson, M., Hoffman, E., Bever, G. S., … and Bhullar, B. A. S. (2017). The skull roof tracks the brain during the evolution and development of reptiles including birds. Nature ecology and evolution, 1(10), 1543-1550.

This is now actually written in our manuscript:

“As is generally the case in non-maniraptoriform dinosaurs (e.g., Witmer and Ridgely, 2008, 2009; Knoll et al., 2015, 2021), many characteristics of the mid- and hindbrain are not perceivable with certainty (however, see Evans, 2005; Morhardt, 2016; Fabbri et al., 2017) on the braincase endocast of DFMMh/FV 581.1 (Figure 1A), which implies scarce correlation of the actual brain and the inner surface of the endocranial cavity (see Watanabe et al., 2019 for ontogenetic variations in recent archosaurs).”

Consideration of the above-mentioned comment:

We included Evans, 2005; Morhardt, 2016 and Fabbri et al., 2017 as suggested by the reviewer. However, Witmer and Ridgely, 2008 state themselves:

“In some archosaurs, such as pterosaurs (Witmer et al. 2003), derived coelurosaurs (Currie 1985, 1995; Currie and Zhao 1993; Osmólska 2004; Kundrát 2007), and perhaps some ornithischians (Evans 2005), the brain resembled that of extant birds in that it largely filled the endocranium to the extent that an endocast fairly represents the general brain structure (Iwaniuk and Nelson 2002). However, the endocast of P. lakustai (Figures 1, 2) is more like that of extant reptiles (and indeed most fossil archosaurs; Hopson 1979; Witmer et al. 2008) in that the endocast is not particularly brain-like in form, suggesting that the brain itself was perhaps markedly smaller than the endocast”

Hence, we consider this paper not suitable for the point of the reviewer as we understand it but we included it in the first bracket of the sentence.

Line 200-205: would it be possible to have a table of comparisons between hearing frequencies between Europasaurus and other sauropodomorphs? This would give a better idea of the peculiarity of Europasaurus in comparison to other sauropodomorphs and will foster a better understanding of the ecology of these animals.

We added the values known from the early diverging sauropodomorph Thecodontosaurus antiquus (Ballell et al., 2021) which are calculated with the same means (Walsh et al., 2009).

Line 320-333: although I agree that modern species are the key to inferring ecological behavior in extinct ones, I would ask the authors to discuss additional scenarios. For example, what if this taxon was simply communicating during the breeding season and was solitary for the majority of its life? Precociality includes this kind of behavior, but this is not discussed at the moment. I would suggest the authors expand this discussion

We added new references, expanded our considerations and hope the reviewer considers this suitable:

“Although the mediolateral width of the lagena does not appear to be associated with auditory capabilities (Walsh et al., 2009), the lagena of Europasaurus is conspicuously thick mediolaterally, especially when compared to its anteroposterior slenderness (Figure 6). […] Together with tropic Late Jurassic conditions in Europe (Armstrong et al., 2016), this may be part of the explanation of the recovered auditory capacities of Europasaurus.”

Figures: these are generally well done. That being said, I would strongly suggest a final figure in the main text showing multiple endocasts and semicircular canals of multiple sauropodomorphs in a phylogenetic context. This will help the general reader to understand the evolutionary trends affecting the evolution of the brain in this clade. I would be happy even with schematic drawings showing multiple taxa.

Such graphics are already present in articles cited by us (Paulina-Carabajal, 2012; Knoll et al., 2012; Martinez et al., 2016; Bronzati et al., 2017; Andrzejewski et al., 2019; Knoll et al., 2019; Ballell et al., 2021; Müller et al., 2021) and we do not see a great benefit of including Europasaurus in such a depiction.